

# Prevalence of chronic conditions in masters games athletes: predictors and comparison to the general population

Fiona Halar[1,2], Helen O'Connor[1,3,†], Mike Climstein[4,5], Tania Prvan[6], Deborah Black[7], Peter Reaburn[8], Wendy Stuart-Smith[3,9], Xiaojing Sharon Wu[10] and Janelle Gifford[1,11]

[1] Sydney School of Health Sciences, Faculty of Medicine and Health, University of Sydney, Camperdown, NSW, Australia
[2] Sydney School of Public Health, Faculty of Medicine and Health, University of Sydney, Camperdown, NSW, Australia
[3] Charles Perkins Centre, University of Sydney, Camperdown, New South Wales, Australia
[4] Physical Activity, Sport, and Exercise Research Group, Faculty of Health Sciences, Southern Cross University, Bilinga, Queensland, Australia
[5] Health & Performance Faculty Research Group, Faculty of Medicine and Health, University of Sydney, Camperdown, New South Wales, Australia
[6] School of Mathematical and Physical Sciences, Faculty of Science and Engineering, Macquarie University, North Ryde, NSW, Australia
[7] Faculty of Medicine and Health, University of Sydney, Camperdown, New South Wales, Australia
[8] Exercise and Sport Science, Faculty of Health Sciences and Medicine, Bond University, Gold Coast, Queensland, Australia
[9] School of Nursing and Midwifery, Faculty of Medicine and Health, University of Sydney, Camperdown, New South Wales, Australia
[10] Department of Nutrition, Faculty of Medical and Health Sciences, University of Auckland, Auckland, New Zealand
[11] Sport and Physical Activity Research and Teaching Network (SPARTAN), University of Sydney, Camperdown, New South Wales, Australia
† Deceased 13th January 2020.

Corresponding author
Janelle Gifford,
janelle.gifford@sydney.edu.au

## ABSTRACT

**Background:** Masters athletes (MA) are typically considered healthier than age-matched non-athletes. However, limited evidence exists on the prevalence of chronic conditions in MA.

**Methods:** Masters athletes competing at the 2017 Australian Masters Games ($n = 4,848$) and 2018 Pan Pacific Masters Games ($n = 14,455$) were invited to complete a survey collecting demographic and health information focused on chronic conditions. Age- and sex-adjusted prevalence of selected chronic conditions in MA was compared with Australian general population data.

**Results:** Overall, 817 MA ($53.7 \pm 10.6$ y, 61% female) completed the survey with 48% reporting ≥1 chronic condition. Cardiovascular conditions were less prevalent in MA *vs.* the general population (11%, 95% CI [9–14%] *vs.* 30%), as were cardiovascular risk factors, anxiety, asthma, cancers, and depression. The prevalence of osteoarthritis in MA was, however, similar (11%, 95% CI [9–14%] *vs.* 14%). Older (>50 yr) *vs.* younger MA were more likely to report osteoarthritis (OR 2.17, 95% CI [1.35–3.48]) and heart conditions (OR 1.85, 95% CI [1.11–3.07]), while younger *vs.* older MA were more likely to report mental health conditions (OR 1.86, 95% CI [1.23–2.82]). Prevalence of mental health conditions was higher in female *vs.* male

MA (17% 95% CI [13–20%] *vs.* 8% 95% CI [5–11%]) and younger *vs.* older MA (18% *vs.* 10%). Employed MA were less likely than MA who were not employed to report having one or more cancers (OR 0.33, 95% CI [0.16–0.69]), cardiovascular conditions (OR 0.47, 95% CI [0.29–0.79]) and hypertension (OR 0.36 95% CI [0.18–0.73]).

**Conclusions:** Cardiovascular and other chronic conditions prevalence was lower in MA compared to age-matched non-athletes, highlighting the value of promoting sport involvement in aging individuals as well as for continuation of participation in younger age groups into MA level. Greater participation of younger and female groups in masters sport to improve mental health, and inclusion of people who are not employed should be supported.

## INTRODUCTION

The financial and social burden of aging populations in developed countries have prompted an interest in promoting healthy and active aging throughout the lifespan. Aging-related research shows that engaging in regular physical activity and sport can improve physical, mental and social wellbeing (*Izquierdo et al., 2025*). In addition, sport participation can provide older adults with an opportunity to be physically active in an enjoyable manner while also reducing the risk of developing chronic conditions (*May, 2018*). Participation in masters sport has increased dramatically over the last three decades, exemplified by participation in the World Masters Games increasing three-fold from 1985 to 2017 (*International Masters Games Association, 2017*).

While age cut-offs vary across competitions and sports, masters athletes (MA) are typically individuals aged 35 years or older who are physically active beyond population guidelines or engage in sporting competitions or systematic training (*Soto-Quijano, 2017*). MA have previously been reported to be healthier and have a lower risk of chronic conditions such as heart disease, hypertension, hypercholesterolemia and type 2 diabetes mellitus than the general population (*Climstein et al., 2023*, *2021*; *Geard et al., 2017*; *May, 2018*). Some researchers have even proposed that MA are exemplars of healthy aging (*Geard et al., 2017*). However, the risk of developing chronic conditions is well recognized as increasing with age (*Chodzko-Zajko et al., 2009*; *Geard et al., 2017*). To date, research examining the prevalence of chronic conditions in MA has been limited to examining specific types of sport, the measurement of general health markers such as body mass index (BMI) or determining the prevalence of a specific chronic condition or small number of chronic conditions (*Climstein et al., 2023*, *2022*; *Fien et al., 2017*; *Gifford et al., 2015*; *Walsh, Heazlewood & Climstein, 2018*).

A more comprehensive empirical investigation of the prevalence of chronic conditions and their possible predictors in MA may offer useful insights in promoting the benefits sports participation to mature-aged and older adults. The primary aim of the present study

was to examine the prevalence of a wide range of chronic conditions in MA and to determine if MA participating at national and international competitions have a lower prevalence of chronic conditions compared to the general Australian adult population. A secondary aim was to understand how key predictors of health (age, sex, socioeconomic and lifestyle risk factors) influenced the prevalence of chronic conditions in MA.

## MATERIALS AND METHODS

A convenience sample of MA were invited to complete an online survey of the 2017 Australian Masters Games (AMG) participants in Tasmania ($n = 4,848$) and 2018 Pan Pacific Masters Games (PPMG) on the Gold Coast ($n = 14,455$), Australia. The minimum age requirement to participate was 18 years (for swimming). Participants were provided with an information sheet and proceeding with the survey indicated consent to participate in the study. PPMG respondents who answered "yes" to having completed the survey at the AMG were exited from the survey before it's commencement. Demographics collected common across both AMG and PPMG for all participants were age, gender, state and postcode (Australia), and country.

The self-report survey collected information on demographics and health, including medically diagnosed chronic conditions. For comparison purposes, the 2017–18 Australian National Health Survey (AHS) conducted throughout Australia from July 2017–June 2018 provided data on the general Australian adult population (20 years of age or older; $n = 17,999,900$). AHS data were obtained *via* the Australian Bureau of Statistics (ABS)/Universities Australia Agreement. The Australian adult population was chosen as the comparison group as the majority (90%) of MA at the games were from Australia.

The primary health outcomes investigated were the presence of chronic conditions, diagnosed by their general practitioner or medical specialist. Similar chronic conditions were grouped by type according to AHS data groupings for comparison: anxiety, asthma, cancers, cardiovascular conditions (includes hypertension), depression, hyperlipidaemia, hypertension, osteoarthritis, osteoporosis, and type 2 diabetes mellitus.

Sociodemographic factors assessed included the age, sex, country of residence, educational attainment, employment status and yearly household income (in Australian dollars) of MA. Health risk factors assessed included body mass index (BMI), smoking status, amount of daily alcohol consumption and physical activity. Physical activity variables included the number of hours per week of cardiovascular-based exercise, number of days per week of resistance-based exercise, number of years previously having competed at a masters level and the sport(s) MA competed in at the games. Registered sports were categorized into the dominant energy system and included: aerobic, anaerobic/power, prolonged high intensity intermittent (PHIT), skill, and mixed (respondents that registered for sports of more than one type of energy system).

Representativeness of responders *vs.* the whole sample was done in an earlier analysis with a chi-squared goodness of fit using SPSS for Windows (V 23.0.0.0; IBM Corp, Armonk, NY, USA). All other statistical analyses were conducted using R (V 4.0.0; R Foundation for Statistical Computing, Vienna, Austria) and a level of significance of $\alpha = 0.05$ was applied *a priori* to determine statistical significance. Descriptive analyses were
conducted overall and analysed by the sex of MA for all sociodemographic and health risk factors. The prevalence of chronic conditions and 95% confidence intervals (95% CI) were estimated overall, by sex and by broad age category (≤50 years and >50 years), the latter stratification was chosen because increase in prevalence for many chronic conditions reported in the AHS such as heart disease, diabetes, osteoarthritis, and osteoporosis increased in the 45–55 age range (*Australian Bureau of Statistics, 2018*) and the middle age was chosen for the cutoff. The prevalence of selected chronic conditions in MA was adjusted for age and sex for comparison with AHS data due to age and sex being more evenly distributed in the AHS data than the data in the current study.

For factors associated with having one or more chronic conditions, multivariable analyses were conducted using logistic regression with purposeful selection of variables (*Fien et al., 2017*). Covariates were selected where there was association between the outcome and the covariate with a $p < 0.25$. Smoking status and registered sports energy system were excluded due to low numbers (<10) in one or more categories of the variable. Univariable analyses were conducted to identify trends where there was insufficient power for multivariable analyses; that is, where the sample size was too small, associations between the outcome and explanatory variables were identified as patterns in the data. Measures of association were provided as odds ratios (OR) with their respective 95% CIs. MA that responded with 'Don't know' to the presence of a chronic condition were excluded from both prevalence and regression analyses.

This study received ethical approval from the University of Sydney Human Ethic Committee (2017/592) and it was supported by AMG and PPMG organizers, who were involved with recruitment and the piloting phase of the survey. Individual MA sports representatives were also involved during the piloting phase. A summary of the results was provided to the Games organizers for dissemination to their games participants.

## RESULTS

A total of 817 MA (response rate = 4.2%) from the AMG ($n = 130$) and PPMG ($n = 687$) were included in the analysis with $n = 28$ (3%) being younger than 35 years, and the majority of these being aged 30 years or over. Females were overrepresented in the sample compared to the proportion of female masters games participants (61% *vs.* 51%, $p < 0.001$), and the mean age of those responding was higher than the mean age of eligible games participants (53.7 ± 10.6 years *vs.* 51.1 ± 10.8 years, $p < 0.001$). The proportion of MA from Australia in the sample was similar to eligible masters games participants across both samples (91% *vs.* 90%, $p = 0.673$).

The characteristics of MA are shown in Table 1. The age of MA ranged from 21 to 87 years across both Games. Male MAs were significantly older than female MAs (55.8 ± 10.6 years *vs.* 52.4 ± 10.4 years, $p < 0.001$) and a higher proportion of male *vs.* female MAs consumed more than two standard drinks per day (9% *vs.* 2%, $p < 0.001$). Female MAs were more highly educated than males (79% *vs.* 70%, $p = 0.007$). More male MAs did either no resistance exercise or three or more days of resistance exercise, and more female MAs did 1–2 days of resistance exercise ($p = 0.018$).

**Table 1 Characteristics of Masters athletes at Masters games in Australia.**

| Characteristics | Male (n = 320) | Female (n = 497) | Total (n = 817) | Sex differences p value[a] |
|---|---|---|---|---|
| Age (years) | 55.8 ± 10.6 | 52.4 ± 10.4 | 53.7 ± 10.6 | <0.001 |
| Australian resident | | | | 0.235 |
| No | 35 (10.9) | 42 (8.5) | 77 (9.4) | |
| Yes | 285 (89.1) | 455 (91.5) | 740 (90.6) | |
| Highest level of education[b] | | | | 0.007 |
| Secondary or Cert I–IV | 94 (29.6) | 105 (21.2) | 199 (24.4) | |
| Tertiary | 224 (70.4) | 391 (78.8) | 615 (75.6) | |
| Employment status[b] | | | | 0.371 |
| Not employed | 67 (21.0) | 91 (18.5) | 158 (19.5) | |
| Employed | 252 (79.0) | 402 (81.5) | 654 (80.5) | |
| Gross household income[c] (per annum) | | | | 0.685 |
| <$60,000 | 50 (15.6) | 87 (17.5) | 137 (16.8) | |
| $60,000–$100,000 | 64 (20.0) | 107 (21.5) | 171 (20.9) | |
| >$100,000 | 154 (48.1) | 218 (43.9) | 372 (45.5) | |
| Not stated or known | 52 (16.2) | 85 (17.1) | 137 (16.8) | |
| BMI (kg/m²)[b] | 27.5 (4.4) | 26.2 (5.3) | 26.7 (5.0) | <0.001[d] |
| BMI classification | | | | <0.001[e] |
| Underweight | 0 (0) | 5 (1.0) | 5 (0.6) | |
| Normal weight | 91 (28.4) | 234 (47.4) | 325 (39.9) | |
| Pre-obesity | 156 (48.8) | 151 (30.6) | 307 (37.7) | |
| Obesity class I | 50 (15.6) | 75 (15.2) | 125 (15.4) | |
| Obesity class II | 17 (5.3) | 18 (3.6) | 35 (4.3) | |
| Obesity class III | 6 (1.9) | 11 (2.2) | 17 (2.1) | |
| Smoking status | | | | 0.773 |
| Never smoked | 214 (66.9) | 325 (65.4) | 539 (66.0) | |
| Ex-smoker | 96 (30.0) | 152 (30.6) | 248 (30.4) | |
| Current smoker | 10 (3.1) | 20 (4.0) | 30 (3.7) | |
| Alcohol consumption[b] | | | | <0.001 |
| 2 or less standards drinks/day | 289 (90.9) | 485 (98.4) | 774 (95.4) | |
| >2 standard drinks/day | 29 (9.1) | 8 (1.6) | 37 (4.6) | |
| Cardio exercise | | | | 0.117 |
| <2.5 h/week | 71 (22.2) | 90 (18.1) | 161 (19.7) | |
| 2.5–5 h/week | 145 (45.3) | 261 (52.5) | 406 (49.7) | |
| >5 h/week | 104 (32.5) | 146 (29.4) | 250 (30.6) | |
| Resistance exercise | | | | 0.018 |
| None | 100 (31.2) | 123 (24.7) | 223 (27.3) | |
| 1–2 days/week | 108 (33.8) | 215 (43.3) | 323 (39.5) | |
| 3 or more days/week | 112 (35.0) | 159 (32.0) | 271 (33.2) | |
| Previously competed at Masters level | | | | 0.292 |
| 2 years or less | 107 (33.4) | 178 (35.8) | 285 (34.9) | |
| 3–10 years | 103 (32.2) | 174 (35.0) | 277 (33.9) | |
| >10 years | 110 (34.4) | 145 (29.2) | 255 (31.2) | |

(Continued)

| Characteristics | Male (n = 320) | Female (n = 497) | Total (n = 817) | Sex differences p value[a] |
|---|---|---|---|---|
| Registered sport(s) energy system | | | | 0.046 |
| Aerobic | 27 (8.4) | 31 (6.2) | 58 (7.1) | |
| Anaerobic/Power | 109 (34.1) | 183 (36.8) | 292 (35.7) | |
| PHIT | 136 (42.5) | 233 (46.9) | 369 (45.2) | |
| Skill | 16 (5.0) | 9 (1.8) | 25 (3.1) | |
| Mixed | 32 (10.0) | 41 (8.2) | 73 (8.9) | |
| Number of chronic conditions | | | | 0.090 |
| No conditions | 150 (46.9) | 206 (41.4) | 372 (45.5) | |
| 1 type of condition | 83 (25.9) | 154 (31.0) | 237 (29.0) | |
| 2 types of conditions | 43 (13.4) | 66 (13.3) | 109 (13.3) | |
| 3 or more types of conditions | 12 (3.8) | 35 (7.0) | 47 (5.8) | |
| Don't know | 32 (10.0) | 36 (7.2) | 52 (6.4) | |

**Notes:**
Data from Masters athletes participating in the 2017 Australian Masters Games and 2018 Pan Pacific Masters Games. Data as Mean ± SD or number (percentage of total). BMI, body mass index; PHIT, Prolonged High Intensity Intermittent; Conditions: arthritic conditions, asthma, bone conditions, cancers (all types), cardiovascular conditions, endocrine or metabolic conditions, food allergies and intolerances, mental health conditions.
[a] p value from Chi-squared test or Independent two sample student t-test, unless stated otherwise.
[b] Missing data excluded for Highest level of education (n = 3; M = 2; F = 1), Employment status (n = 5; M = 1; F = 4), BMI and BMI Classification (n = 3; M = 0; F = 3) and Alcohol consumption (n = 6; M = 2; F = 4).
[c] Australian dollars.
[d] Statistically significant, but not considered clinically significant.
[e] p value from Fisher's Exact Test.

**Table 2 Prevalence of chronic conditions in Masters athletes at Masters games in Australia.**

| Chronic condition | Total (Yes/No) responses | Male prevalence % (95% CI) | Female prevalence % (95% CI) | Total prevalence % (95% CI) |
|---|---|---|---|---|
| Arthritic conditions | 801 | 17.6 [13.5–22.2] | 15.0 [11.9–18.4] | 16.0 [13.5–18.7] |
| Gout | 814 | 2.2 [0.9–4.5] | 0.4 [0–1.4] | 1.1 [0.5–2.1] |
| Osteoarthritis | 805 | 12.4 [9.0–16.5] | 13.9 [10.9–17.3] | 13.3 [11–15.8] |
| Rheumatoid arthritis | 815 | 3.4 [1.7–6.1] | 1.2 [0.4–2.6] | 2.1 [1.2–3.3] |
| Asthma | 810 | 8.2 [5.4–11.8] | 11.6 [8.9–14.7] | 10.2 [8.2–12.5] |
| Bone conditions | 799 | 1.3 [0.3–3.2] | 6.4 [4.4–9.0] | 4.4 [3.1–6.0] |
| Osteopenia | 804 | 0.3 [0–1.7] | 3.5 [2.1–5.6] | 2.2 [1.3–3.5] |
| Osteoporosis | 811 | 0.9 [0.2–2.7] | 3.0 [1.7–5.0] | 2.2 [1.3–3.5] |
| Cancers (all types) | 800 | 5.1 [2.9–8.2] | 3.3 [1.9–5.3] | 4.0 [2.8–5.6] |
| Cardiovascular conditions | 804 | 11.9 [8.6–16.0] | 9.9 [7.4–12.9] | 10.7 [8.6–13.0] |
| Hypertension | 809 | 6.0 [3.6–9.1] | 3.5 [2.0–5.5] | 4.4 [3.1–6.1] |
| Heart diseases | 812 | 0.9 [0.2–2.7] | 0.2 [0–1.1] | 0.5 [0.1–1.3] |
| Other cardiovascular conditions | 808 | 7.9 [5.2–11.4] | 7.6 [5.4–10.3] | 7.7 [5.9–9.7] |
| Endocrine or metabolic conditions | 813 | 5.0 [2.9–8.1] | 3.0 [1.7–4.9] | 3.8 [2.6–5.4] |
| Hyperlipidaemia | 816 | 1.9 [0.7–4.0] | 1.6 [0.7–3.2] | 1.7 [0.9–2.9] |
| Type 2 diabetes mellitus | 815 | 2.2 [0.9–4.5] | 0.6 [0.1–1.8] | 1.2 [0.6–2.2] |
| Other types of diabetes | 814 | 0.9 [0.2–2.7] | 1.0 [0.3–2.3] | 1.0 [0.4–1.9] |

| Chronic condition | Total (Yes/No) responses | Male prevalence % (95% CI) | Female prevalence % (95% CI) | Total prevalence % (95% CI) |
|---|---|---|---|---|
| Food allergies or intolerances | 797 | 9.0 [6.1–12.7] | 17.7 [14.4–21.4] | 14.3 [11.9–16.9] |
| Mental health conditions | 798 | 7.7 [5–11.3] | 16.6 [13.4–20.2] | 13.2 [10.9–15.7] |
| Anxiety related problems | 803 | 5.7 [3.4–8.9] | 10.9 [8.2–14.0] | 8.8 [7.0–11.0] |
| Depression | 799 | 3.5 [1.8–6.2] | 9.0 [6.6–11.9] | 6.9 [5.2–8.9] |
| Other mental health conditions | 805 | 1.3 [0.3–3.2] | 1.0 [0.3–2.4] | 1.1 [0.5–2.1] |

**Note:**

Data from Masters athletes (MA) participating in the 2017 Australian Masters Games and 2018 Pan Pacific Masters Games ($n$ = 817). Data as percentage of total (95% confidence interval of percentage). Having a type of chronic condition (*e.g.*, mental health condition) was coded as 'Don't know' if the response to one related type of condition (*e.g.*, anxiety) was 'Don't know' and another related type of condition (*e.g.*, depression) was 'No', therefore proportions may not add up to 100%. MA that answered 'Don't know' excluded. Heart diseases: angina, heart attack, heart failure; Other cardiovascular conditions: rheumatic heart disease, stroke, hypotension, atherosclerosis, oedema, irregular heartbeats/tachycardia, cardiac murmurs/heart valve disorders, haemorrhoids, varicose veins. Other types of diabetes: Type 1 diabetes mellitus, pre-diabetes (IFG, IGT), type unknown; Other mental health conditions: other mood (affective) disorders, mild cognitive impairment.

The prevalence of different chronic conditions in MA is shown in Table 2. Approximately one half (48%, $n$ = 393) of MA reported having one or more chronic conditions. Arthritic conditions were the most prevalent (16%, 95% CI [14–19%]) followed by food allergies and intolerances (14%, 95% CI [12–17%]), mental health conditions (13%, 95% CI [11–16%]), cardiovascular conditions (11%, 95% CI [9–13%]), asthma (10%, 95% CI [8–13%]), bone conditions (4%, 95% CI [3–6%]), cancers (4%, 95% CI [3–6%]) and endocrine and metabolic conditions (4%, 95% CI [3–5%]).

Arthritic conditions were the most prevalent in male MAs (18%, 95% CI [14–22%]) followed by cardiovascular conditions (12%, 95% CI [9–16%]) and food allergies and intolerances (9%, 95% CI [6–13%]). The least prevalent among male MAs were bone conditions (1%, 95% CI [0–3%]) and endocrine and metabolic conditions (5%, 95% CI [3–8%]). In female MAs, food allergies and intolerances were most prevalent (18% 95% CI [14–21%]) followed by mental health conditions (17%, 95% CI [13–20%]). The least prevalent among female MAs were endocrine and metabolic conditions (3%, 95% CI [2–5%]) and cancers (3%, 95% CI [2–5%]). Depression (9% 95% CI [7–12%] *vs.* 4% 95% CI [2–6%]) and osteopenia (4% 95% CI [2–7%] *vs.* 0% 95% CI [0–2%]) were more prevalent in female *vs.* male MA.

Figure 1 shows the MA age- and sex-adjusted prevalence of selected chronic conditions compared to the general Australian adult population. For all chronic conditions, except osteoarthritis, the prevalence was significantly lower in MA than the general population. The largest difference in prevalence between MA and the general population was for cardiovascular conditions (11%, 95% CI [9–14%] *vs.* 30%), followed by hypertension (5%, 95% CI [3–6%] *vs.* 19%), asthma (10%, 95% CI [7–12%] *vs.* 21%) and hyperlipidaemia (2%, 95% CI [1–3%] *vs.* 12%).

Female MAs were more likely than males to have chronic conditions (OR 1.69 95% CI [1.24–2.31]). Masters athletes who reported consuming more than two standard alcohol drinks per day were more likely to have chronic conditions than MA consuming two or less standard alcoholic drinks per day (OR 2.14 95% CI [1.06–4.51]).

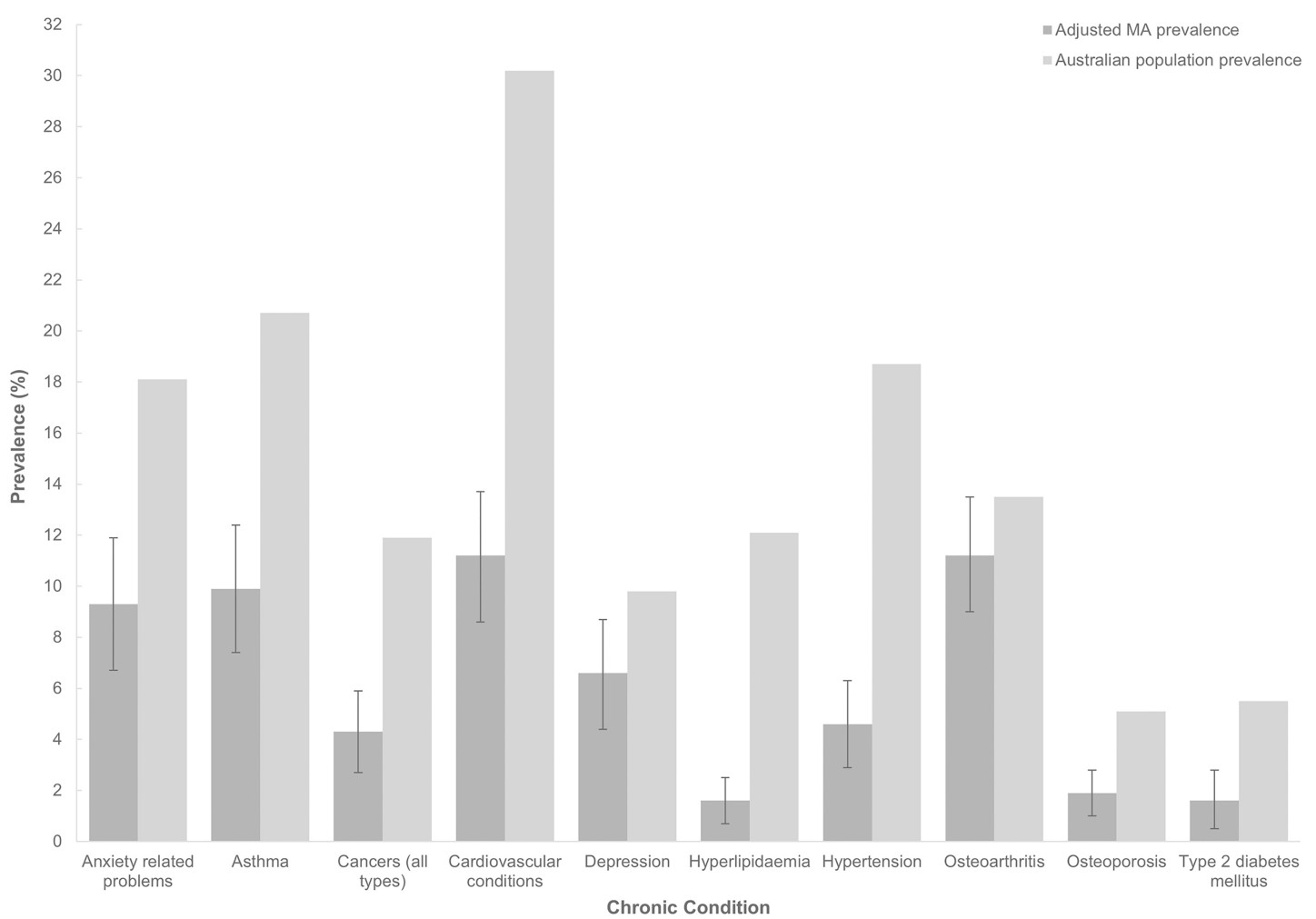

**Figure 1 Comparison of the prevalence of chronic conditions between Masters athletes at Masters games in Australia and the general Australian adult population.** Masters athletes (MA) data from MA participating in the 2017 Australian Masters Games and 2018 Pan Pacific Masters Games ($n = 817$). Population data based on the 2017–18 Australian National Health Survey (AHS; 20 years old or more; $n = 17,999,900$). MA data as percentage of total (95% confidence interval of percentage). MA prevalence age-sex-adjusted using the AHS as the target population. Three age groups used in the adjustment (20–39, 40–59, 60+). MA that responded 'Don't know' for each condition excluded (anxiety related problems, $n = 12$; asthma, $n = 9$; cancers, $n = 18$; cardiovascular conditions, $n = 11$; depression, $n = 18$; hypertension, $n = 8$; osteoarthritis, $n = 11$; osteoporosis, $n = 4$; type 2 diabetes mellitus, $n = 4$). Cardiovascular conditions: hypertension, heart diseases, rheumatic heart disease, stroke, hypotension, atherosclerosis, oedema, irregular heartbeats/tachycardia, cardiac murmurs/heart valve disorders, haemorrhoids, varicose veins.

Being a female MA was associated with a higher likelihood of having anxiety (OR 1.99, 95% CI [1.16–3.56]), depression (OR 2.68, 95% CI [1.41–5.55]) and food allergies and intolerances (OR 2.14, 95% CI [1.38–3.42]) than male MA. Within the MA cohort as a whole, increasing age was associated with having one or more cancers (OR 1.06, 95% CI [1.03–1.10]), cardiovascular conditions (OR 1.04, 95% CI [1.02–1.07]), hyperlipidaemia (OR 1.07, 95% CI [1.02–1.12]), hypertension (OR 1.04, 95% CI [1.01–1.08]), osteoarthritis (OR 1.04, 95% CI [1.02–1.06]), osteoporosis (OR 1.12, 95% CI [1.07–1.18]) and type 2 diabetes mellitus (OR 1.10, 95% CI [1.04–1.17]). Increasing age was inversely associated with anxiety (OR 0.95, 95% CI [0.93–0.98]).

Tertiary educated MA were more likely than MA educated at a secondary or vocational training certificate level to have one or more cardiovascular conditions (OR 2.04, 95% CI [1.12–4.04]). Employed MA were less likely than MA not employed to report having one or more cancers (OR 0.33, 95% CI [0.16–0.69]), cardiovascular conditions (OR 0.47, 95% CI [0.29–0.79]) and hypertension (OR 0.36 95% CI [0.18–0.73]). A yearly household income of more than $100,000 was protective of having osteoarthritis in MA compared to a yearly household income of less than $60,000 (OR 0.56 95% CI [0.32–0.98]).

A higher BMI was associated with having anxiety (OR 1.05, 95% CI [1.01–1.10]), depression (OR 1.06, 95% CI [1.01–1.11]), hypertension (OR 1.12, 95% CI [1.06–1.18]) and osteoarthritis (OR 1.05, 95% CI [1.01–1.09]). Masters athletes previously having competed at a masters level for 3 to 10 years were less likely to have anxiety than those previously having competed for 2 years or less (OR 0.58 95% CI [0.35–0.93]).

Older (>50) *vs.* younger MA were more likely to report arthritis (OR, 2.23, 95% CI [1.44–3.45]), specifically osteoarthritis (OR 2.17, 95% CI [1.35–3.48]) and having a heart condition (OR 1.85, 95% CI [1.11–3.07]). Younger *vs.* older MA were more likely to report mental health conditions (OR 1.86, 95% CI [1.23–2.82]), specifically depression (OR 1.81, 95% CI [1.04–3.14]) and anxiety (OR 1.94, 95% CI [1.19–3.17]).

## DISCUSSION

Masters athletes have been suggested as examples of healthy aging as they are physically active and typically reported to be healthier than age-matched non-athletes (*Geard et al., 2017*; *May, 2018*). The present findings strongly suggest that MA have a lower prevalence of most chronic conditions compared to the general Australian adult population, highlighting the potential health benefits of participation in sport for mature-aged and older adults.

In the current study, arthritic conditions (gout, osteoarthritis and rheumatoid) were observed to be the most prevalent type of chronic condition reported in MA (16%), with the most common of these being osteoarthritis. *Climstein et al. (2011)* reported a similar prevalence of arthritis (osteoarthritis and rheumatoid) in veteran rugby athletes (15%). However, another previous study examining North American MA competing at Masters games found a lower prevalence (10%) of arthritis (osteoarthritis and rheumatoid) (*DeBeliso et al., 2014*). Multiple factors are associated with the risk of developing osteoarthritis. These include person-related (*e.g.*, age, sex, BMI, and bone mineral density), abnormal joints and alignment, prior joint injuries, and high impact sports (*e.g.*, running, soccer, football, rugby, and tennis), which may account for the difference in prevalence in MA from the different cohorts (*Buckwalter & Martin, 2004*; *Neogi & Zhang, 2013*; *Soto-Quijano, 2017*). As expected, the prevalence of osteoarthritis was higher in older MA than in younger MA.

Food allergies and intolerances were observed in MA in the present study (14%). However, accurate data on food allergies are difficult to obtain due to differences in methodologies used for data collection, with a lower prevalence indicated when objective testing is used *vs.* questionnaires (*Wing-Kin Wong, 2024*). Research using objective measures (electronic health records) suggests that the actual prevalence of food allergies

and intolerance is much lower in adults (4%) (*Acker et al., 2017*). The MA in the current study may have reported having undiagnosed food allergies and intolerances, which may have inflated prevalence numbers. Similar to *Acker et al. (2017)*, female MA in the present study reported having a higher prevalence of food allergies and intolerance compared to males (18% *vs.* 9%).

A study on North American MA participating in masters games found the prevalence of depression to be similar (5%) to MA in the current study (7%) (*DeBeliso et al., 2014*). This is further supported by a larger scale study on MA (*n* = 8,072) which reported the mean incidence of depression as 6.3% (39% of male and 61% of female MA) (*International Masters Games Association, 2017*; *Walsh, Heazlewood & Climstein, 2018*). The present data indicates that mental health conditions, including depression and anxiety, were more likely to be reported by female than male MA (17% *vs.* 8%), MA in the younger age range (18% *vs.* 10% for older MA), and MA who had previously competed for 2 years or less. Globally, females are more likely to experience depression and anxiety conditions than males (*GBD 2019 Mental Disorders Collaborators, 2022*). *Albert (2015)* suggests that hormonal fluctuations related to the menstrual cycle, postpartum and during perimenopause influence the occurrence of depression. The cut-off for young MA in our cohort was 50, around the time of menopause (*Riecher-Rössler, 2020*). Therefore, it is likely that the higher prevalence of mental health conditions in young female MA in this study is partly related to these physiological changes before and during the menopausal transition. The poorer mental health in younger MA in the current study may be related to the "career-and-care crunch" described by *Mehta et al. (2020)*. The period between 30 and 45 years of age (established adulthood) may be a time where life changes related to the establishment of relationships and family, the need to financially and physically care for them, and the increasing potential that ageing parents may need more care may occur. Epidemiological studies investigating the relationship between physical activity and mental health have shown that regular physical activity is associated with a decreased risk for depression or anxiety in adults (*Bangsbo et al., 2019*; *Chodzko-Zajko et al., 2009*). Other evidence also suggests that participation in masters-level sport may improve mental health beyond that of general physical activity (*Geard et al., 2017*). We therefore suggest that further support to assist female and younger established adultgroups to be involved and stay in masters sport to improve mental health is warranted.

In the current investigation, the largest difference in prevalence between MA and the general population was for cardiovascular conditions, with MA also having a significantly lower prevalence of cardiovascular risk factors (hypertension, hyperlipidaemia, type 2 diabetes) (*Climstein et al., 2023*, *2022*, *2018*). Cardiovascular conditions are the leading causes of mortality globally, with the prevalence of both cardiovascular conditions and risk factors known to increase with age (*North & Sinclair, 2012*). The preventative and therapeutic benefits of regular physical activity related to cardiovascular conditions are well established, with research showing a decrease in the relative risk of cardiovascular mortality among more cardiovascular fit persons compared with moderately fit or sedentary age-matched peers (*Bangsbo et al., 2019*). Muscular strength and power have

also been shown to influence cardiovascular mortality independent of cardiovascular fitness (*Chodzko-Zajko et al., 2009*).

MA are a diverse group for many reasons. However, as a cohort, they are similar in terms of their higher socioeconomic status (*Dionigi, 2016*). The majority of MA in the present study were highly educated (tertiary level), employed, and in a higher household income bracket (>$AUD100,000 *per annum*). The current study also found that employed MA (*vs.* not employed) were less likely to have one or more chronic conditions. Previous research has shown that socioeconomic indicators, such as income, wealth, education, and employment are known to be important determinants of health (*Cockerham, Hamby & Oates, 2017*). Moreover, other research conducted in adults has found socioeconomic status to be related to both overall physical activity and leisure-time physical activity, and education level and being employed linked to the amount of exercise or sport adults undertake (*O'Donoghue et al., 2018*). In the current study some participants were not seeking work (*e.g.*, retired, homemaker), therefore not being employed may not be a good indicator of socioeconomic status.

Lifestyle factors such as smoking, alcohol consumption, physical inactivity, and unhealthy diet along with metabolic risk factors such as overweight or obesity, are well known risk factors for chronic conditions (*Izquierdo et al., 2025*). In the present study, the proportion of MA that were current smokers or consumed more than two standard alcoholic drinks per day was low compared to the general Australian population (4% and 5% respectively). However, MA consuming more than two standard alcoholic drinks per day were more likely to have one or more chronic conditions than MA consuming two or less standard alcoholic drinks per day. Physical inactivity is associated with an increased risk of chronic conditions in older adults (*Bangsbo et al., 2019*; *Chodzko-Zajko et al., 2009*). Although it seems paradoxical that MA may be inactive, designated time for sport and physical activity, and other activities of daily living such as work does not preclude sedentary behaviour. In fact, *Júdice et al. (2022)* found that total sedentary behaviour and screen time measured in 135 national and international athletes were positively associated with percentage of fat mass end negatively associate with percentage of fat free mass. To our knowledge there are no studies that specifically measure sedentary behaviour in MA. In terms of diet, a recent systematic review found that MA consumed a higher energy diet with more micronutrients compared the general population; the authors speculated that the greater energy budget afforded by higher levels of physical activity likely contributing to a better quality diet (*Guo et al., 2023*).

On the whole, it appears that the standard lifestyle risk factors may somewhat mitigate the risk of chronic conditions in MA. Regularly participating in exercise or sport can also significantly reduce these risks (*Bangsbo et al., 2019*; *Chodzko-Zajko et al., 2009*). The majority of MA in the present study met or exceeded Australian physical activity guidelines (*Australian Government Department of Health, 2014*). In concordance with previous research in MA, the proportion of overweight or obese MA in our group was lower than the general Australian adult population (60% *vs.* 67%) (*Australian Bureau of Statistics, 2018*; *Fien et al., 2017*). *Walsh, Heazlewood & Climstein (2018)* found that in most, but not all studies on MA, BMI was lower than control subjects. Given the link

between overweight/obesity with chronic conditions, it appears the lower BMI and increased physical activity levels in the MA in the present study may be major contributors to the lower levels of cardiovascular disease and risk factors observed in the MA.

The present data suggest that female MA are more likely than males to have one or more chronic conditions. This finding is similar to the general Australian population, where a higher proportion of females than males aged 15 to 64 years report one or more chronic conditions, and a higher proportion of females than males 65 years and older had two or more chronic conditions (*Australian Bureau of Statistics, 2018*). These findings may be the result of primary care consultation rates being lower among males, which may translate into less diagnosed chronic conditions in males compared to females (*Wang et al., 2013*). However, cohort studies indicate the previously discussed lifestyle factors (*Shang et al., 2020*) as well as inadequate sleep and stress play a role in multimorbidity (presence of two or more conditions) (*Niebuur et al., 2023*; *Du et al., 2024*) in both men and women. Sleep and stress were not measured in our MA but the lack of research in the area presents an opportunity for further investigation.

The present investigation examined the prevalence of a wide range of chronic conditions in MA, some of which have not previously been investigated in this cohort. Widely used predictors of health were also examined across a wide range of factors, and the sample size was larger than most previous studies examining chronic conditions in MA. However, there was a lack of adequate power for some analyses due to the low prevalence of some health risk factors (*e.g.*, smoking) and chronic conditions. Other limitations included a low response rate, results being based on self-reported data, and potentially healthier individuals competing at masters games, all of which may lead to response bias. There was also a higher representation of females and a potential bias towards individuals who are interested in health and participating in surveys, although these issues are common in health surveys.

## CONCLUSIONS

To lower the financial and social health burden of an aging population, governments, health agencies, sporting organisations, and health professionals need to understand and promote healthy and active aging throughout the lifespan. The present study provided comprehensive evidence that MA have a lower prevalence of most chronic conditions compared to age-matched non-athletes in the general population. It provides strong support for health promotion efforts both for encouraging sports participation into older age and transitioning younger athletes into masters sport to maintain their health. Greater participation of younger and female groups in masters sport to improve mental health, and inclusion of people who are not employed and from lower socioeconomic groups should be supported. Further exploration of successful treatment strategies for chronic conditions in MA and diet and physical activity behaviors is warranted.

## ACKNOWLEDGEMENTS

The authors would like to thank the study participants for their involvement in the study the Masters games organizers (especially Dee Kapene and Cameron Hart) and our

colleagues at the University of Tasmania (especially Lyndal Bond, Sandy Murray, and Dr Simone Lees) for their support of the study.

### Funding

The senior author, Associate Professor Janelle Gifford, received a Britton Craigie Scholarship from the School of Health Sciences (University of Sydney) to enable attendance at the Australian Masters Games in 2017 for the first data collection included in this manuscript. The funders had no role in study design, data collection and analysis, decision to publish, or preparation of the manuscript.

### Grant Disclosures

The following grant information was disclosed by the authors:
Britton Craigie Scholarship from the School of Health Sciences (University of Sydney).

### Competing Interests

Mike Climstein is a Section Editor for PeerJ (Sports Medicine and Rehabilitation).

### Author Contributions

- Fiona Halar analyzed the data, prepared figures and/or tables, authored or reviewed drafts of the article, and approved the final draft.
- Helen O'Connor conceived and designed the experiments, authored or reviewed drafts of the article, and approved the final draft.
- Mike Climstein conceived and designed the experiments, authored or reviewed drafts of the article, and approved the final draft.
- Tania Prvan analyzed the data, authored or reviewed drafts of the article, and approved the final draft.
- Deborah Black analyzed the data, authored or reviewed drafts of the article, and approved the final draft.
- Peter Reaburn conceived and designed the experiments, authored or reviewed drafts of the article, and approved the final draft.
- Wendy Stuart-Smith conceived and designed the experiments, authored or reviewed drafts of the article, and approved the final draft.
- Xiaojing Sharon Wu performed the experiments, analyzed the data, authored or reviewed drafts of the article, and approved the final draft.
- Janelle Gifford conceived and designed the experiments, performed the experiments, analyzed the data, authored or reviewed drafts of the article, and approved the final draft.

## Human Ethics

The following information was supplied relating to ethical approvals (*i.e.*, approving body and any reference numbers):

The University of Sydney Human Ethics Committee approved the study (Protocol number: 2017/592).

## Data Availability

Data is available at Open Science Framework:

Gifford, Janelle, Fiona Halar, Helen O'Connor, Mike Climstein, Tania Prvan, deborah black, Wendy Stuart-Smith, Peter Reaburn, and Xiaojing Wu. 2024. "Prevalence of Chronic Conditions in Masters Athletes." OSF. December 22. DOI: 10.17605/OSF.IO/YSKRD.

## Supplemental Information

Supplemental information for this article can be found online at http://dx.doi.org/10.7717/peerj.18912#supplemental-information.

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
