# Peer review of "Prevalence of chronic conditions in masters games athletes: predictors and comparison to the general population"

_PeerJ, doi:10.7717/peerj.18912_

## Round 0.1 · original submission · Major Revisions

· Academic Editor

Major Revisions

Dear Authors

Two experts in the study field have reviewed the manuscript. One of the reviewers (R1) raised critical points regarding the methodological concerns. We invite you to submit a revised version of the manuscript that addresses the points raised by the reviewers.

We look forward to receiving your revised manuscript.

Best regards

Yung-Sheng Chen, Ph.D.
Academic Editor

Reviewer 1 ·

Basic reporting

no comment.

Experimental design

In the Materials & Methods section:
1. What was the study's participation rate? Did individuals that excluded from participating differ demographically (education, income, health, etc.) from study participants?
2. It is unclear the statistical method used in the analysis.
3. It is unclear how the covariates were selected in the final model analysis.
4. There is no definition of the measurement of the prevalence of chronic conditions. Is it point prevalence or period prevalence?
5. Line 134, why is stratified by 50 years old?
6. Line 142-143, what is the sentence meaning?

Validity of the findings

In the Results section:
1. Fig.1 shows the MA age- and sex-adjusted prevalence of selected chronic conditions compared to the general Australian adult population. Were the age- and sex-adjusted prevalence of selected chronic conditions calculated in the general Australian adult population? The different definitions of prevalence in these two populations. How did the authors compare?
2. No table has been provided for the results of logistic regression.
3. There are no results to validate the secondary aim of this study.

In the Discussion and Conclusion section:
1. The results revealed that younger versus older MA were more likely to report mental health conditions. Please discuss the findings.
2. Some important factors associated with chronic conditions, such as dietary habits, were not included in this study. The authors did not further discuss this part.

Additional comments

This study has many problems for the study design and methods to be improved for publication.

Reviewer 2 ·

Basic reporting

Language and Clarity: The manuscript is well-written, with clear and professional English throughout.
Background and Context: The introduction provides sufficient background and context, outlining the relevance of the research within the broader field of aging and sports participation. Relevant prior literature is cited appropriately.
Article Structure: The article follows an appropriate structure with well-organized sections (e.g., Introduction, Methods, Results, and Discussion). Figures and tables are relevant, clearly labeled, and contribute effectively to the narrative.
Raw Data: The manuscript indicates that data has been shared according to PeerJ's policy, though a detailed check of data accessibility should be confirmed.

Experimental design

Relevance and Scope: The research question is clearly defined and meaningful, addressing a knowledge gap in the literature regarding chronic condition prevalence among Masters athletes.
Ethics and Standards: The investigation appears to have been conducted to a high ethical standard, with proper approval from the University of Sydney Human Ethics Committee.
Methodology: The methods are described in sufficient detail, allowing reproducibility. The design, including the use of logistic regression and appropriate statistical measures, ensures robustness.

Validity of the findings

Data and Analysis: The data provided are robust, statistically sound, and controlled, with comprehensive analysis including odds ratios and confidence intervals.
Conclusions: The conclusions are well-linked to the research questions, highlighting both the prevalence of chronic conditions and the health benefits of sports participation. They are appropriately limited to the results, avoiding unsupported causal claims.

Additional comments

This manuscript is a valuable contribution to the literature on chronic conditions in Masters athletes and the comparison to the general population. The study is comprehensive, well-structured, and presents its findings in a clear manner. With minor amendments to ensure clarity and a thorough check of data accessibility, I recommend that this manuscript be accepted for publication, with minor amendments.

Suggested Minor Amendments:

1. Correct referencing format: in some instances, the author's initial is included in the in-text citations. 2. 2.Ensure all citations conform to APA standards by using only the author's last name and year.
3.Review and update references: some references, such as "World Health Organisation 2010," appear outdated. Consider including more recent references where possible.
4.Line 79 & 89: Update the citation to include: "Climstein, M., Walsh, J., DeBeliso, M., Heazlewood, T., Sevene, T., Del Vecchio, L., & Adams, K. (2023). Resting blood pressure in master athletes: immune from hypertension? Sports, 11(4), 85."
5. Correct typographical errors: remove the incorrect symbol and replace it with P=0.05.
4. Line 238: Address citation error by adding a publication date after "Climstein and Colleagues."
5. Line 279: Add a full stop after the parentheses to ensure proper punctuation.

---

## Round 0.2 · accepted · Accept

· Academic Editor

Accept

Dear Authors,

I would like to express my big heart for your patience and efforts to improve the quality of the manuscript. Your submission is now endorsed by two experts for acceptance of publication in PeerJ. Congratulation!!!

Thank you for submitting your article to PeerJ. I look forward to receiving your research and review articles in the future.

Best Regards
Ph.D. Yung-Sheng Chen

Reviewer 1 ·

Basic reporting

The authors have taken on board the review comments and have taken care to revise the manuscript.

Experimental design

no comment.

Validity of the findings

no comment.

Additional comments

no comment.

Reviewer 2 ·

Basic reporting

improved

Experimental design

n/a

Validity of the findings

n/a

Additional comments

The manuscript has been significantly improved, making it a valuable contribution to the field